# Community Health Workers as a Strategy to Tackle Psychosocial Suffering Due to Physical Distancing: A Randomized Controlled Trial

**DOI:** 10.3390/ijerph18063097

**Published:** 2021-03-17

**Authors:** Dorien Vanden Bossche, Susan Lagaert, Sara Willems, Peter Decat

**Affiliations:** Department of Public Health and Primary Care, Faculty of Medicine and Health Sciences, Ghent University, 9000 Ghent, Belgium; Susan.Lagaert@UGent.be (S.L.); Sara.Willems@ugent.be (S.W.); peter.decat@ugent.be (P.D.)

**Keywords:** community health workers, primary healthcare, mental health, psychosocial support, vulnerable populations, COVID-19, health crisis

## Abstract

Background: During the COVID-19 pandemic, many primary care professionals were overburdened and experienced difficulties reaching vulnerable patients and meeting the increased need for psychosocial support. This randomized controlled trial (RCT) tested whether a primary healthcare (PHC) based community health worker (CHW) intervention could tackle psychosocial suffering due to physical distancing measures in patients with limited social networks. Methods: CHWs provided 8 weeks of tailored psychosocial support to the intervention group. Control group patients received ‘care as usual’. The impact on feelings of emotional support, social isolation, social participation, anxiety and fear of COVID-19 were measured longitudinally using a face-to-face survey to determine their mean change from baseline. Self-rated change in psychosocial health at 8 weeks was determined. Results: We failed to find a significant effect of the intervention on the prespecified psychosocial health measures. However, the intervention did lead to significant improvement in self-rated change in psychosocial health. Conclusions: This study confirms partially the existing evidence on the effectiveness of CHW interventions as a strategy to address mental health in PHC in a COVID context. Further research is needed to elaborate the implementation of CHWs in PHC to reach vulnerable populations during and after health crises.

## 1. Introduction

Since March 2020, the world is facing a global public health crisis, as the coronavirus disease 2019 (COVID-19) emerged as a menacing pandemic. Besides the rising number of cases and fatalities as a consequence of this pandemic, there has also been significant socioeconomic, political, and psychosocial impact [1]. Billions of people are quarantined in their own homes as nations have locked down to implement physical distancing as a measure to contain the spread of infection [1]. Physical distancing and lockdown measures, work disruptions, and school closures, have suddenly changed social life and daily routines. A major effect of these measures has been the reduction of social contacts, with a consequent increase in social isolation and feelings of loneliness, which are associated with increased anxiety, depression, and suicidal behavior [2,3,4,5,6,7,8,9]. Multiple lines of evidence confirm these profound psychosocial effects of the COVID-19 pandemic and physical distancing measures [7,8,9,10]. The psychological sequelae of the pandemic will probably persist for months and years to come. Some groups may be more vulnerable than others to the psychosocial effects of pandemics. In particular, people with a psychiatric history, a precarious social context, a limited network, an uncertain residence status, old age, chronic illness, or going through a recent critical event such as bereavement and divorce are at increased risk for adverse psychosocial outcomes [9]. COVID-19 will disproportionately affect vulnerable populations, worsening prevailing inequalities [3,7,11].

Primary care professionals, such as family physicians, are key figures in this COVID-19 pandemic [12,13]. Primary healthcare (PHC) is the first point of contact for patients with symptoms, worries, anxiety, and questions concerning the pandemic. Additionally, family physicians and other primary care professionals have an important role in addressing the emotional and psychosocial outcomes as part of the pandemic response [14]. This COVID-19 outbreak is a challenge for each of the family physician’s core competencies, as they are described in the European definition of General Practice [15,16]. Primary care management requires solutions to tackle the increased number of patient contacts and to separate COVID and non-COVID flows. Person-centered care needs to be maintained in the shift to telephone consultations. Decision-making skills must account for the changed epidemiology and the need for regular and COVID-related care. A comprehensive approach includes COVID-specific risk management and health education. Community orientation is evidently extremely important in the context of an infectious outbreak. Containing the spread of infection on the one hand and making sure vulnerable and frail patients are not left behind on the other hand are both community responsibilities in which PHC practices and primary care professionals play major roles. Finally, psychological, sociocultural, and existential dimensions define the holistic context in which the family physician and the primary care team operate [14]. However, many primary care professionals are overburdened and cannot adequately reach their vulnerable patients to meet their increased need for psychosocial support [12,13].

A potential answer to the task overload of family physicians and other primary care professionals during the COVID-19 crisis could be ‘task shifting’ being the “rational redistribution of tasks among health workforce teams” [17]. Specific tasks are delegated, if appropriate, from highly specialized health workers to less specialized health workers in order to make more efficient use of human resources, and certain health worker tasks are moved to members of the community [18]. In 1978, the WHO conference on PHC at Alma Ata explicitly cited community health workers (CHWs) as being one of the cornerstones of comprehensive PHC. CHWs are members of the community where they work, are supported by the health system but are not necessarily a part of its organization, and have shorter training than professional workers [19]. They provide basic health services, and contribute to achieving the key principles of community health and PHC: equity, community involvement, responding to local health needs, and inter-sectoral collaboration. The concept of a ‘task shifting approach’ reinforced the role of CHWs.

CHWs have played task-shifting roles in the global health arena for decades. The history of CHWs traces back to the 1970s and their introduction principally aimed to improve maternal and child health, and the management of common infectious diseases in settings with limited health workforce and low access to basic health services notably in low-and middle-income countries (LMICs) [20]. More recently, the use of CHWs has attracted attention in some high-income countries (HICs) where, despite the more developed health systems, large inequities in healthcare access and outcomes amongst different population groups can be noted [21]. The growing interest in CHWs in HICs is also being driven by concerns about shortage in health workforce, and the escalating burden of chronic and complex diseases that is driving a significant increase in health services demand and costs in many developed countries [21]. Most literature on CHWs in HICs comes from the United States and shows the significant role that CHWs play in engaging with patients and families and helping them to navigate the complex health and social systems by providing culturally appropriate care, health education, and advocacy [22]. This results in positive health outcomes in population groups experiencing disadvantage such as migrants and low socio-economic communities [23,24,25,26], increased access and utilization of PHC services, reduced hospital admissions, and improved post-hospital care [27,28].

Given the growing evidence that CHWs are effective in improving physical health outcomes, increased attention has been focused on incorporating CHWs into mental health services [18,29,30]. In LMIC settings, evidence for the effectiveness of task sharing in mental health care and CHW-delivered mental health support and care exists across a continuum of roles and tasks, for a range of mental health-related problems and disorders, particularly for common mental disorders [29,31,32,33,34]. Evidence from a recent systematic review from the United States [30] shows that CHW models of mental health service delivery can be effective in addressing disparities in care for underserved populations, as two-thirds of the randomized controlled trials (RCTs) included in the systematic review demonstrated positive mental health outcomes for traditionally underserved communities over a comparison condition [30]. Given the strong impact of culture and other social determinants on psychological wellbeing [18,29,35], CHWs seem to be in an excellent position to address the most vulnerable ones and to provide psychosocial support to people who are experiencing challenging circumstances possibly impacting their mental health [18,36,37,38,39,40,41,42]. Economic analyses associated with CHW-supported mental health initiatives have provided evidence that CHW-delivered mental health care is cost-effective [29]. As the evidence base for the effectiveness of mental health care by CHWs has grown, increased attention has been paid to various approaches needed for effective implementation of CHW’s service delivery in HIC settings. However, beyond the sphere of research, the actual uptake of the practice of task sharing of mental health care by CHWs at a PHC level in HICs has been limited [29,43]. To our knowledge, this RCT is one of the first studies in European PHC context to test a CHW intervention in the area of mental health.

Despite the importance of CHWs as a task shifting strategy on one hand and in extending health services to vulnerable populations filling health system gaps on the other hand, CHWs are often under-utilized in the acute response to infectious disease outbreaks and additional roles for CHWs in promoting pandemic preparedness exist [11,44,45]. The proposal to offer a fast response by engaging CHWs to support citizens has been frequently suggested in this current COVID-19 health crisis [45]. In Ghent, Belgium, a successful pilot project of CHWs has been running since the beginning of 2019. As a response to many primary care professionals’ concerns on the actual and longer-term mental health of their vulnerable patients, the framework for CHWs’ roles and responsibilities of this existing project was broadened and further implemented as a strategy to offer psychosocial support to vulnerable people who are at risk to become victims of fear and social isolation in these challenging times.

This article aims to evaluate the effect of a CHW intervention on psychosocial suffering among patients with a limited social network during the COVID-19 pandemic. More specifically, this study aimed to test the intervention’s effect on different psychosocial outcomes (i.e., emotional support, social isolation, social participation, anxiety and fear of COVID-19) on one hand and the intervention’s effect on self-rated change in psychosocial health on the other hand. We hypothesized that, compared with patients receiving ‘care as usual’, patients receiving the intervention would have better psychosocial outcomes (including increased experience of emotional support and social participation and lower social isolation, anxiety and fear of COVID-19) and a positive change in patients’ self-rated psychosocial health state. By modeling and testing of this CHW intervention, we discuss how the cadre for CHWs’ roles and responsibilities may be engaged to potentially improve pandemic and community-level resilience.

## 2. Materials and Methods

This study is a community-based, open label, two-arm, parallel-group, randomized clinical trial with equal allocation of participants between the intervention and the control arms. This trial is registered (ClinicalTrials.gov Identifier: NCT04426305) and approved by the ethical committee of Ghent University Hospital, Belgium (BC-07744). All participants provided written informed consent. The trial was conducted for five months, including baseline and follow-up outcome measures. The trial is part of a realist evaluation of this CHW intervention.

### 2.1. Study Setting

This study was conducted at Ghent University. Patient recruitment and rollout of the intervention took place in the city of Ghent. In Ghent, CHWs are active since the beginning of 2019. The CHWs are volunteers who, from their background or experience, are more aware of the problems of people in a vulnerable context. After training and under supervision, they take on the following tasks: to detect problems and to inform and advise, support, stimulate, and empower vulnerable patients. In this study, the existing practice of working with CHWs was further rolled out to support people at increased risk of psychosocial suffering through the physical distancing measures because of COVID-19.

### 2.2. Participants

Eligible patients (1) had a limited social network; (2) were older than 18 years; (3) had a psychiatric history, or a precarious social context, or an uncertain residence status, or a chronic illness, or were going through a recent critical event such as bereavement or divorce, or were older than 65 years; (4) had a score of ≤7 on the screening questions for emotional support (“I have people who care about what happens to me” and “There are people I can talk to”) and ≥7 on the screening questions for anxiety (“In the past 7 days I felt fearful” and “In the past 7 days I felt uneasy”), with scoring options for each screening question being 1 = Never; 2 = Rarely; 3 = Sometimes; 4 = Usually; 5 = Always. Exclusion criteria adopted in this trial were: (1) having serious psychiatric problems in the current medical history, such as schizophrenia, substance abuse, depression with suicidality etc.; (2) not being fluent in Dutch, French, English, Spanish, Turkish, or Arabic; (3) having symptoms of possible COVID-19 infection; (4) being pregnant.

### 2.3. Patient Enrollment/Recruitment Procedures

Patients were selected following a two-step approach. In a first step, a convenience sample of 11 PHC practices was selected by contacting first all practices located in one of the deprived areas of Ghent, followed by a snowball technique to find additional practices until the sample of 21 PHC practices was reached. In a second step, the participating practices selected a sample of patients to participate in the study. Hereto all patients visiting the practice between April and June 2020 and complying the inclusion criteria were invited to participate in the study. Additionally physicians actively identified eligible patients from their patient files and actively contacted them to participate in the study. When patients consented to participate, their contact details were passed on to the researchers. All patients were first contacted by phone by the researchers for check-in and exclusion criteria. Patients were then randomized into the control or the intervention group. Patients were recruited until the prespecified sample size target was reached. Follow-up surveys were completed on 8 August 2020.

### 2.4. Procedures and Data Collection

Randomization into intervention and control group was done by the researchers of the research team using a simple randomization technique, i.e., flipping a coin. Owing to the nature of the study, blinding of treatment was not possible for participants or researchers. All participants were asked to fill in a written questionnaire twice (at baseline and 8 weeks later). These questionnaires were developed by the researchers of the research team of Ghent University (specified below) and aimed to measure the influence over time of corona measures on psychosocial well-being. The questionnaires consisted of questions assessing the following areas: (1) outcome measures; (2) process measures and; (3) sociodemographic data (complete questionnaires available in Appendix A). Patients filled in this questionnaire in a face-to-face meeting with a research assistant at home or by phone. If patients consented for this, this information was shared with their family physicians so a follow-up could be guaranteed. The intervention for patients randomized to the corresponding group was initiated in the period of two weeks after the baseline questionnaire was filled in. Control group patients received ‘care as usual’, which means that these patients were approached by their caregivers, either by telephone or during an encounter in practice, asking how they were doing during lockdown measures. In addition, (telephone) advice according to the physical distancing measures and tips for handling feelings of loneliness were given to patients who indicated that they were experiencing mental difficulties.

### 2.5. Intervention

The intervention was set up in collaboration with the Department of Welfare and Health of the city of Ghent. CHWs for this study were recruited from the pool of CHWs already working in the city of Ghent. As well, new CHWs were recruited via an open call on the local online volunteering platform and from other community projects in the city of Ghent. All candidates participated in two online training modules of 2 h. This training was developed by the coordinators of the CHW project in Ghent and by the researchers of the research team. The modules entailed communication skills, providing correct information, recognizing alarming signals presented by patients and safety measures to prevent COVID-19 infection. During the intervention, on-demand support was provided if needed by the CHWs and intervision and peer-to-peer coaching were provided in small groups once a month.

CHWs provided 8 weeks of hands-on, tailored support to patients spanning the domains of social support, coaching, advocacy, and navigation to healthcare if needed. The overall goal was to offer presence to patients who were socially isolated or who felt lonely or anxious. By being present, CHWs offered a sympathetic ear and gave attention to their patients’ worries, stories, and questions. CHWs were also instructed to check whether their patients were correctly informed about the most recent distancing measures. If this was not the case, the CHWs provided and explained the updated preventive measures. Moreover, when patients presented with alarming signals according to their psychosocial state, CHWs took responsibility to inform their patients’ caretakers and the coordinating team about the situation. As general goal setting to acquire intervention standardization was required, the CHWs were asked to aim for a total of 8 contacts over a period of 8 weeks. After the matchmaking between CHW and patient was done by the project coordinators, the CHWs received their patients’ contact details. The first contact was always made by phone. In this first contact, CHWs presented themselves, checked-in with how their patients were doing, and explored how their assigned patients wanted to organize the next contacts. Further on, the CHWs communicated with patients at regular basis, depending on expressed needs of patients. The content of the contacts could vary from sending text messages, WhatsApp messages, e-mails or postcards, over doing Skype meetings or phone calls, to going for regular walks in the park. The time of a contact could vary from a few minutes to two hours. After approximately 8 contacts, the CHWs were instructed to announce to their patients that they were going to pause or stop the contact because the study period was going to an end, that a researcher of the research team would come by to take the follow-up questionnaire, and that after that they could of course again stay in touch with a CHW if desired. CHWs did not directly provide health education or clinical care, and when these needs arose, CHWs navigated patients to the appropriate healthcare provider. If a patient was hospitalized during the intervention, CHWs continued contacting the patient unless patients expressed no further need to this. Intervention guidelines are codified in the form of detailed manuals, in-person and online training, documentation, and reporting.

### 2.6. Measures

#### 2.6.1. Sociodemographic Data

To determine the general characteristics of our study population, questions on the following sociodemographic data were asked: (1) sex, (2) age, (3), highest degree achieved, (4) current work or activity, (5) migration background, and (6) ethnicity (Appendix A). More sociodemographic variables were asked in the questionnaires, e.g., whether the individual lived alone, whether they were actually shielding due to physical health difficulties, whether they had support at home, and whether they had recent preexisting psychological complaints. Those were not further elaborated since the aim of this trial was to evaluate the impact of the intervention on feelings of loneliness; so the goal was not to describe the influencing factors of loneliness.

#### 2.6.2. Outcome Measures

The prespecified primary outcomes were the mean change in feelings of emotional support, social isolation, ability to participate in social roles and activities, and anxiety. These outcomes were measured at baseline and 8 weeks later using the Patient-Reported Outcomes Measurement Information System (PROMIS™) [46] validated short forms for these prementioned topics and with adding of relevant and specific questions from the specified items banks. Each question has five response options ranging in value from one to five. For all PROMIS instruments, applies that the higher the score, the higher the occurrence of the construct measured. PROMIS instruments are scored using item-level calibrations. This means that the most accurate way to score a PROMIS instrument is to use the Health Measures Scoring Service or a data collection tool that automatically calculates scores. PROMIS instruments are expressed in a standardized T-score. The average score in a population is assigned 50. The standard deviation is equal to 10 points. PROMIS uses the scores of the calibration sample on which the item bank was created to calculate T-scores. In this case, this is a sample from the general U.S. population. These approaches to scoring employ a common, highly accurate method that uses each participant’s responses to each item administered. We refer to this method as “response pattern scoring”. The specific PROMIS questions for each outcome topic (i.e., emotional support, social isolation, ability to participate in social roles and activities, and anxiety) are available for each outcome measure in the complete questionnaires in Appendix A.

For the ‘fear of COVID-19′ primary outcome, a new scale was developed consisting of 9 items, measured continuously, with scores ranging from 0 to 10. The development of items was based on relevant issues observed in patient encounters during the first rise in COVID-19 incidence in Belgium (Appendix A). For this self-developed fear of COVID-19 scale, the researchers of the research team conducted a principal component analysis to evaluate whether questions capture multiple components. The screeplot of the eigenvalues indicated that only one factor (the only factor with an eigenvalue clearly exceeding 1), on which all 9 questions of the fear of COVID-19 scale have high loadings, should be retained. Reliability analysis of a mean scale based in the 9 questions of the COVID-19 scale shows very good internal consistency (Cronbach’s α = 0.871).

#### 2.6.3. Process Measures

Process measures were the self-rated change in emotional support, social isolation, social participation, anxiety and fear of COVID-19, using the Patient Global Impression of Change (PGIC) scale. This is a continuous scale ranging from −5 to +5, where a score of −5 means ‘very much worse’, 0 means ‘unchanged’, and +5 means ‘very much better’. The questions of this scale were asked in the post-intervention questionnaire (i.e., after 8 weeks).

In the intervention group additionally, a set of questions on the satisfaction with the intervention were added as a part of the post-intervention questionnaire. These questions were scored using a continuous scale with scores ranging from 0 to 10.

Principal component analyses of the PGIC and satisfaction scales showed presence of one component. Reliability analyses of the PGIC and satisfaction scales showed Cronbach’s alpha values of 0.558 and 0.752 respectively. The low value for the PGIC scale could be explained by the low item number of this scale. The specific questions of both PGIC and satisfaction scales are available in the full questionnaire in Appendix A.

### 2.7. Statistical Analysis

Data were analyzed from August 2020 to November 2020.

Sample sizes were based on detecting a between-arm difference in mean change from baseline in PROMIS emotional support, social isolation, ability to participate in social roles and activities, and anxiety T-scores at 8 weeks of 5 points, the meaningful change for this instrument [46,47]. To achieve at least 80% power using an independent samples *t*-test at a two-sided significance level 5%, assuming a common standard deviation of 9.65 points in both groups, we required 60 participants per arm. To account for 7% attrition, we aimed to accrue 130 participants in total.

Statistical analyses were performed using the statistical package IBM SPSS Statistics software (SPSS) version 27 (SPSS Inc., Chicago, IL, USA). Descriptive comparisons between group baseline characteristics were performed with χ^2^ or Fisher’s exact tests for categorical variables and with independent-samples t-tests and Wilcoxon rank sum tests for continuous variables. The effectiveness of the intervention was evaluated using linear mixed models (LMMs) [48]. These models accounted for the repeated measurements study design (baseline measurement and after 8 weeks) with an unstructured covariance matrix. LMMs were fitted with group (intervention versus control), time (at 8 weeks versus baseline), and their interaction group x time as fixed factors. Results were expressed as estimated marginal means with corresponding 95% CIs. Comparisons were reported in terms of expected baseline-adjusted mean differences between groups at 8 weeks (group x time interaction) with 95% CIs. All 135 randomized patients were included in the intention-to-treat analysis. Mean difference in PGIC at 8 weeks was analyzed with an independent-samples t-test. This was a complete case analysis which would only yield valid results when missing PGIC data were missing completely at random.

All hypothesis tests were performed two-sided at the 5% significance level, corresponding 95% confidence intervals were given. No adjustment for multiple testing was made.

## 3. Results

### 3.1. Study Patients

In total, 21 PHC practices expressed their interest to involve CHWs in their care for patients. They identified 191 patients for whom they saw a role for the CHW (as presented in Figure 1). Thirty-two (16.8%) of them were not eligible for the study because they did not meet the inclusion criterium of having a score of ≤7 on the screening questions for emotional support and ≥7 on the screening questions for anxiety, posed on the first contact by phone by the researchers. The patients who did not meet this criterium, and so had high scores on questions on feelings of emotional support and had low scores on questions on feelings of anxiety, were seen as not having a potential benefit of the intervention. For this reason, they were not eligible. Of the remaining 159 eligible patients, 135 (84.9%) provided informed consent, and 24 (15.1%) declined to participate right before the first questionnaire (as presented in Figure 2). Reasons for declining were lacking the time to enroll, not wanting a CHW, and not wanting to participate in research after having time to consider.

Complete primary outcome data were available in nearly equal numbers in both study groups at 8 weeks (58 [85.3%] vs. 54 [80.6%]; *p* = 0.47). Reasons for drop-outs were refusing to fill in the second questionnaire (because of lack of time, not wanting to participate further in research) and early disruption of the CHW intervention.

Sociodemographic descriptors for the participating patients are presented in Table 1. Participants were 135 patients living in Ghent and known by a family physician working in the same urban area. Participants ranged in age from 19 to 93, with a mean age of 60.04. A majority of the participants were female (62.2%) and 37.8% were male. A chi-square test revealed no statistically significant difference in sex ratio between the two research groups (*p* = 0.076). Educational degree and economic activity were approximately equally distributed over their respective categories and in both study groups (as presented in Table 1). In addition, 32.6% of participants had a background of migration and 80.7% of participants were living in Belgium for more than 10 years.

### 3.2. Outcome Measures

Independent-samples *t*-test results revealed no statistically significant difference between emotional support, social isolation, and ability to participate in social roles and activities scores of the two groups before the intervention (*p*  >  0.05). For anxiety and fear of COVID-19 scores however, independent-samples *t*-test results did reveal a statistically significant difference (*p* = 0.049 and *p* = 0.017, respectively), despite random attribution to intervention and control group. For these scores, the patients of the intervention group tend to have higher scores (meaning higher levels of anxiety and fear of COVID-19) than the control group patients at baseline (as presented in Table 2).

A LMM was built to determine a statistically significant difference between randomization groups on the outcome variables (emotional support, social isolation, social participation, anxiety and fear of COVID-19) controlling for baseline scores on these outcome variables. Preliminary checks were conducted to ensure that there was no violation of the assumption of normality.

We found no significant between-group difference in mean change from baseline in emotional support, social isolation, ability to participate in social roles and activities, anxiety and fear of COVID-19. (Table 3 and Table 4). In addition, the estimated mean differences in change from baseline at 8 weeks between the two groups were also not clinically relevant. The 95% confidence intervals fell entirely between the margins of meaningful change of [−5%, +5%], except for anxiety where the mean decrease in anxiety in the control group might be larger than in the intervention group.

### 3.3. Process Measures

In total, 50 volunteering CHWs were trained to provide an intervention and 67 pairs of CHW-patient were matched. Eighty-one percent of patients assigned to a CHW engaged with the program for the full 8 weeks. The remaining 13 patients (19%) were drop-outs; as they decided that they no longer wanted to participate in the research project or they did not want to be contacted by a CHW anymore because they felt no need to.

Independent-samples *t*-test analysis showed a significant difference between intervention and control group in the mean score on the PGIC scale (*p* = 0.027, 95%CI [−0.81; −0.05]). Patients in the intervention group reported a positive change in self-rated emotional support, social isolation, social participation, anxiety and fear of COVID-19, whereas the control group patients on average reported no change of status of these outcomes (Table 5).

The mean score on the satisfaction scale for intervention group patients (*n* = 53) was 8.01 (SD 1.75), indicating very high personal satisfaction with the intervention and likelihood to repeat and recommend this intervention.

## 4. Discussion

This RCT tested whether a PHC-based CHW intervention could tackle psychosocial suffering due to physical distancing measures. We failed to find a significant effect of the intervention on experiences of emotional support, social isolation, ability to participate in social roles and activities, anxiety and fear of COVID-19. However, the intervention did lead to significant improvement in self-rated impression of change in psychosocial health.

This study showed that it is possible to engage a pool of volunteers in a short period to alleviate the acute need for psychosocial support in PHC practice. This is in line with the findings of two recent reviews on CHWs and mental health, which described the added value of CHWs’ commitment to alleviate the mental healthcare burden in HIC settings, particularly given evidence of feasibility and acceptability with underserved populations [18,29]. More specifically, review findings indicate that mental health CHWs are acceptable to patients, as evidenced by low attrition and high intervention attendance. In this study, acceptability is reflected in the very high satisfaction scores of intervention group patients. In addition, CHWs ability to liaise closely with family physicians and other primary care professionals, identifying problems early, and supporting patient follow-up indicate potential to reduce unnecessary workload burden on primary care professionals, improving access while reducing use of acute and secondary care services [28].

This trial was a natural experiment, pragmatically probed on existing and pressing needs in actual PHC practice. This focus on implementation in an actual and real setting is a major strength of our trial and increases validity and generalizability of our findings. An additional strength is the high response rate for this type of research, which illustrates the actual existence of the explained psychosocial needs in our vulnerable target population. Another strength of our trial is that it is a randomized controlled design, which allows for causal conclusions concerning the intervention’s effect on self-rated psychosocial health.

This trial has limitations. First, we do not know if the effects persisted beyond the 8 weeks of the trial. Second, RAs filling in the questionnaires with the patients were not blinded. Randomization was applied on the moment of the first questionnaire to make it possible for the RAs to ask informed consent for the intervention in the intervention group after the questionnaire was done. This could possibly create an observer bias. We considered it unethical to randomize after the first questionnaire was completed, because we did not want to create false expectations of a treatment for vulnerable patients in challenging circumstances, who could—at the end—be randomized in a control group. Third, since the PGIC process measures are self-reported changes, we cannot rule out the placebo effect of self-rating. Finally, as with all patient-level trials, selection effects may bias the results because participants may differ from those who decline to participate.

Several reasons might explain why this CHW intervention did not reach its intended outcomes. First, the program might be too complex to be implemented and evaluated within an 8-week time frame. Second, the implementation of the intervention might have been suboptimal in some cases. Launching a tailored intervention in a short time in the prevailing exceptional circumstances was a major challenge. The recruitment and online training of CHWs, the screening and interviewing of eligible patients, the matchmaking between CHWs and patients, and the organization of the intervention and follow-up were all set up in extreme short time span. This could possibly have an influence on both CHWs and patients, who were sometimes overwhelmed. As an attempt to reach intervention standardization in this RCT, we proposed that CHWs endeavored for a total of 8 contacts over a period of 8 weeks. On the other hand, this requirement could possibly encumber tailoring of the intervention to patients’ needs and CHWs’ context and impact spontaneity of the contact, which is known as a major asset of CHW strategies. This difficult balance between standardization and customization with room for spontaneity is a known challenge in testing CHW interventions [43]. Third, the intervention’s content, the target population, and the outcome measures might not match perfectly. The intervention included a total of 8 contacts over a period of 8 weeks, in which CHWs offered presence, gave the right information on the pandemic’s physical distancing measures, and were gatekeepers for patients who could not bear the situation anymore. By contrast, we tested PROMIS-scores on emotional support, social isolation, social participation, anxiety and fear of COVID-19. As we know, a support network is an important factor in building resilience, although more is needed and over a longer period of time to change people’s mental status in a sustainable way. The depth of the psychosocial suffering for this vulnerable study patient population is fundamental. Extra psychosocial suffering was recently added, due to the lockdown and physical distancing measures. In that sense, this intervention cannot eliminate the pre-existing (fundamental) suffering, but it could show an effect in the perception of psychosocial support and on feelings of satisfaction during these extraordinary circumstances. So, on a more superficial level, we did show an effect on psychosocial suffering due to lockdown and physical distancing measures (i.e., self-rated impression of change measured by PGIC). The PROMIS measures, on the other hand, assess the fundamental feelings of emotional support, social isolation, social participation, anxiety and fear of COVID-19, which are too deeply rooted in this vulnerable patient population to expect them to be changed after an 8-weeks intervention by a CHW. Taking a closer look at our vulnerable study population, it is also possible that the expectation to see an improvement that exceeds the meaningful change target might be too high and in that way not adapted to the target population. Possibly these outcome measures might be not sensitive enough for the vulnerable study population.

An interesting finding in this trial is the discrepancy between the significant improvement in self-rated impression of change in psychosocial health and the very high satisfaction scores of intervention group patients versus the findings of no statistical significance for the outcome measures. As in this study, descriptive evaluations of practice models to embed CHWs into PHC often find that recipients are appreciative of the service, but this does not always translate into measurable differences in outcomes. These contradicting findings require further research and as mentioned before, will be further explored through realist evaluation methodology. A hypothesis possibly explaining this contradiction could be that the intervention on itself was not community-oriented in its design nor in its implementation. The determination of the target population in this study, i.e., patients with limited social network, was done by the researchers of the research team. To reach this target population, we relied on family physicians and other primary care professionals to identify patients for whom they saw a role for the CHW. The determination of the suitable target population and the selection of participants were therefore done from a top-down approach and did not evolve from an exploration of patients’ expressed needs. This outreaching way of recruitment of participants could be seen as a valuable strategy to meet the most vulnerable patients, but could also mean that we did not include the most suitable patients for this intervention and therefore could not show a statistically significant effect. Additionally, in order to optimize the potential of CHW strategies, it could be argued that this CHW-workforce needs to be accessible to the entire community and not just those that have been identified with a very specific health need. In general, interventions by CHWs show beneficial results, particularly when these CHWs are integrated in the PHC system and part of a multidisciplinary PHC team [43]. In our Belgian setting, within this pilot project operating in Ghent, CHWs are not deployed in the systematic way that may be needed to fully exploit their potential. They operate in parallel to PHC services and although they may signpost and refer to PHC practices, they generally do not work in or with PHC practices directly. So although CHWs may have access to some detailed and nuanced health ‘intelligence’ gleaned by interacting with individuals, households, and communities, this cannot be easily captured or used by family physicians and other primary care professionals. This lack of integration with existing primary care will inevitably lead to missed opportunities, inefficiencies, and duplication [28,49].

Taken together, we believe there is sufficient research base and a plausible case to further evaluate the use of CHWs integrated into primary care in Belgium. The promising role for CHWs as a strategy to reach out to vulnerable communities, to identify problems early, and to support patient follow-up indicates potential to reduce unnecessary workload burden on primary care professionals. It is important to gain more insight in the working elements of CHW strategies to improve community-oriented care in PHC practice. The results of the realist evaluation will probably lead to insights in what works for whom and under which circumstances. An important condition for implementation in PHC practice is that CHWs are integrated in primary care teams. Future research should therefore take a closer look at this organizational embeddedness of CHW strategies in the primary care team. If this further research demonstrates the benefits we postulate, then there would be a good case for scaling up this approach in a HIC setting.

## 5. Conclusions

This study confirms partially the existing evidence on the effectiveness of CHW interventions as a strategy to address mental health in PHC in a COVID context. Although we failed to find a significant effect of the intervention on feelings of emotional support, social isolation, social participation, anxiety and fear of COVID-19, we can conclude that intervention group patients’ perception of their psychosocial health has positively changed and that they were highly satisfied about the intervention. Our findings support the potential of CHW interventions as a task shifting strategy to reduce family physicians’ and other primary care professionals’ workload. Future research should focus on the implementation of CHW interventions in PHC settings. More specifically, insights in working elements could enable us to develop strategies for community-oriented intervention design and for integration in the primary care team, which could lead to a more profound evidence base for implementation of CHW interventions in HIC PHC settings.

## Figures and Tables

**Figure 1 ijerph-18-03097-f001:**
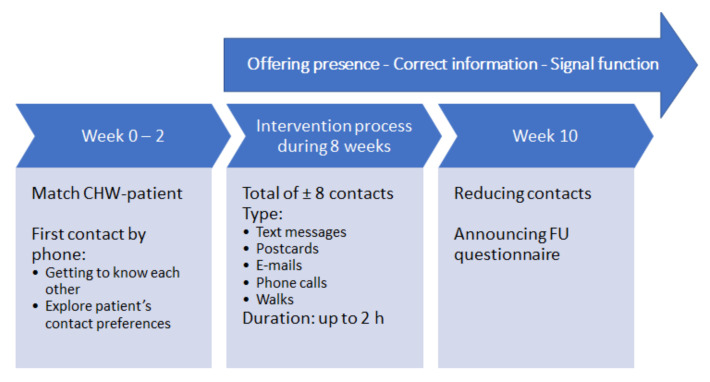
Summary of the intervention process.

**Figure 2 ijerph-18-03097-f002:**
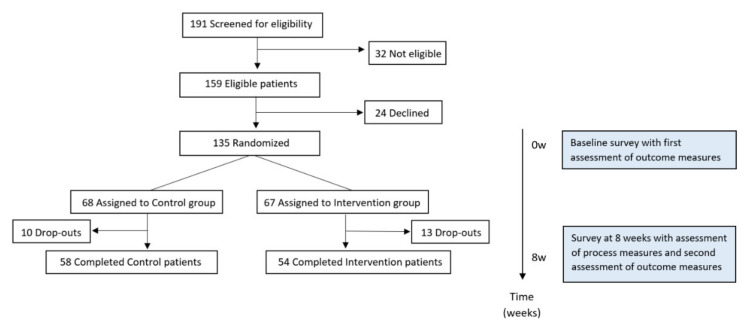
Diagram showing flow of patients through the clinical trial.

**Table 1 ijerph-18-03097-t001:** Sociodemographic and health characteristics for 135 participants.

		Randomization			
		Control		Intervention	
		N	%	N	%
Sex	Male	31	45.6%	20	29.9%
	Female	37	54.4%	47	70.1%
Age (yrs)	<25	4	5.9%	2	3.0%
	25–39	8	11.8%	12	17.9%
	40–64	24	35.3%	20	29.9%
	≥65	32	47.1%	33	49.3%
Highest degree achieved	No or primary school	12	17.6%	13	19.4%
	Primary secondary education	20	29.4%	15	22.4%
	Higher secondary education	23	33.8%	20	29.9%
	Higher education	13	19.1%	19	28.4%
Work or activity	Student	3	5.1%	1	1.6%
	Worker/Servant/Self-employed	7	11.9%	7	11.5%
	Job-seeking	3	5.1%	4	6.6%
	Houseman/housewife	1	1.7%	0	0.0%
	Retired	30	50.8%	37	60.7%
	Disability	11	18.6%	9	14.8%
	Integration course	1	1.7%	2	3.3%
	Other	3	5.1%	1	1.6%
Migration background	No	45	66.2%	46	68.7%
	Yes	23	33.8%	21	31.3%
Time in Belgium (yrs)	<1	4	6.0%	5	7.5%
	1–5	4	6.0%	3	4.5%
	6–10	5	7.5%	4	6.0%
	>10	54	80.6%	55	82.1%
Region of origin	European Region	50	73.5%	53	79.1%
	African Region	6	8.8%	8	11.9%
	Eastern Mediterranean Region	8	11.8%	5	7.5%
	Region of the Americas	2	2.9%	1	1.5%
	South-East Asia Region	2	2.9%	0	0.0%

**Table 2 ijerph-18-03097-t002:** Comparison of the two research groups, control (*n* = 68) and intervention (*n* = 67), in terms of outcome measures at baseline.

Outcome Variable	Group	Mean	Std. Deviation
Emotional support T-score	Control	45.43	8.77
	Intervention	43.41	7.75
Social isolation T-score	Control	53.49	8.66
	Intervention	56.41	8.86
Social participation T-score	Control	44.40	8.26
	Intervention	44.58	8.11
Anxiety T-score	Control	59.42	9.91
	Intervention	62.55	8.34
Fear of COVID-19 Mean scale	Control	4.07	2.35
	Intervention	5.01	2.17

**Table 3 ijerph-18-03097-t003:** Mean scores for the outcome measures of the intention-to-treat analysis (*n* = 135).

Outcome Variable	Baseline Score						Postintervention Score at 8 Weeks					
	Control			Intervention			Control			Intervention		
	Mean	95% CI		Mean	95% CI		Mean	95% CI		Mean	95% CI	
		Upper	Lower		Upper	Lower		Upper	Lower		Upper	Lower
Emotional support	45.43	43.44	47.42	43.41	41.40	45.41	45.44	43.34	47.53	44.09	42.02	46.17
Social isolation	53.49	51.39	55.59	56.41	54.29	58.52	51.83	49.56	54.11	53.70	51.45	55.95
Social participation	44.40	42.44	46.36	44.58	42.60	46.56	46.74	44.45	49.03	46.68	44.41	48.95
Anxiety	59.42	57.22	61.62	62.55	60.34	64.77	53.61	50.67	56.56	59.06	56.16	61.96
Fear of COVID-19	4.07	3.52	4.61	5.01	4.46	5.56	3.57	3.01	4.14	4.01	3.44	4.57

**Table 4 ijerph-18-03097-t004:** Adjusted mean differences for the outcome measures of the intention-to-treat analysis (*n* = 135).

Outcome Variable	Baseline-Adjusted Mean Difference between Groups Postintervention (95% CI)	*p*-Value for Interaction
Emotional support	0.68 (−2.24 to 3.59)	0.647
Social isolation	−1.04 (−4.21 to 2.14)	0.520
Social participation	−0.24 (−3.21 to 2.72)	0.870
Anxiety	2.32 (−1.89 to 6.52)	0.278
Fear of COVID-19	−0.51 (−1.12 to 0.10)	0.103

**Table 5 ijerph-18-03097-t005:** Patient Global Impression of Change (PGIC) process measure independent-samples *t*-test results.

Process Measure	Group	Mean	Std. Dev.	Independent-Samples *t*-Test		
				*p*-Value	95% CI	
					Lower	Upper
PGIC Mean Scale	Control (*n* = 59)	0.01	0.90	0.027	−0.81	−0.05
	Intervention (*n* = 61)	0.43	1.18			

## Data Availability

The data presented in this study are available on request from the corresponding author. The data are not publicly available due to ethical considerations and privacy restrictions.

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
