# Peer review of "Community Health Workers as a Strategy to Tackle Psychosocial Suffering Due to Physical Distancing: A Randomized Controlled Trial"

_ijerph, 2021, doi:10.3390/ijerph18063097_

Round 1

Reviewer 1 Report

The paper entitled "Community health workers as a strategy to tackle psychosocial suffering due to physical distancing: a randomized controlled trial" aim to test whether a primary healthcare (PHC) based community health worker (CHW) intervention could tackle psychosocial suffering in patients with limited social networks. Intervention group received an 8 weeks tailored psychosocial support to the intervention group. Control group patients received ‘care as usual’. However, authors failed to find a significant effect of the intervention on the psychosocial health measures.

My concern is both, with the outcome measures and duration of the intervention. This is not a double blinded RCT, therefore, self-report measures are questionable. In addition, an 8 weeks assessment seems to me very short to detect significant changes. The final sample size seems relatively small as well, considering drop-outs.  

The overall goal of the paper is well-intended, but the contribution to the paper  to the field is scarce as it reads right now. 

Reviewer 2 Report

Dear authors and Editors.

I am uncertain if the attached file shows for you. Therefore, I am inserting all comments also in this field. 

Introducing comment

I want to thank the authors and the Editor of International Journal of Environmental Research and Public Health for the opportunity to review this interesting manuscript entitled “Community health workers as a strategy to tackle psychosocial suffering due to physical distancing: a randomized controlled trial”. This is a very interesting and important manuscript.

Below are my suggestions divided into the sections in which they appear, with reference to either page or sub-section of the paper.

Abstract:

No comment.

Introduction:

  1. On page 1 the authors state “Multiple lines of evidence confirm these profound psychosocial effects of the COVID-19 pandemic and physical distancing measures” [10]. Are there more references available to support the formulation “multiple lines of evidence”? In the reference number 10 for example the authors suggest two additional references to support this statement. Otherwise, perhaps rephrase the beginning of this sentence.

  1. General comment: In the introduction on page 2 (and discussion), the authors have chosen to focus on family physicians. The introduction does not include other primary care professionals such as for example nurses. I wonder if the authors have considered including also nurses in the introduction alternatively talking more broadly about “primary health care staff” in the introduction? When reading the introduction I reflected upon that for example nurses also meet with this patient group and may also have been overburdened and feel that they cannot adequately meet the needs of their vulnerable patients (and thereby may also be helped by CHWs). The reference nr 13 used in the introduction also talks more generally about “health care staff” and not only physicians. I see later in the methods that the physicians had an important role in the recruitment of the participants so perhaps that is the motivation of the focus.

  1. The introduction to the field of CHW is very interesting. The authors may want to consider if it could be slightly shortened by for example shortening the sections referring back to CHW in LMIC, alternatively the development of CHW in HICs. This to highlight the role of CHW in this specific paper. I.e. CHW in relation to psychosocial suffering/mental health (services).

  1. On page 3, the authors state “Economic analyses associated with these initiatives have provided evidence that CHW-delivered mental health care is cost-effective” [29]. It would be helpful if the authors could clarify what is included in “these initiatives”.

  1. On page 3, the authors state “In Ghent, Belgium, a successful pilot project of CHWs has been running since the beginning of 2019”. Is it possible to add a reference to this project, also to support that the pilot project was successful?

Methods:

  1. On the top of page 4, the authors describe that the current study is a part of a realist evaluation. Is there any reference to the larger study?

  1. Under 2.2 Participants. In relation to the selection criteria of participants. 1 Was the “precarious social context” and the “uncertain residence status” defined in any way? Was the time-period for the “recent” critical event defined?

  1. Page 4. Under heading 2.4. Would suggest the authors to consider revising the sentence “Roughly, the questionnaires consisted of questions assessing…” to “The questionnaires consisted of questions assessing the following areas…”

  1. General question: On what basis were the eight weeks determined to be the time-period for the intervention?

  1. Page 5. Under heading 2.5. Check spelling of the word “parc”. Should it be “park”?

  1. Where there any information collected on the proportion of type of contacts? I.e it would be interesting to know if the majority of contacts were via text (messages/mails/post cards), phone or physical contact (walks in parks). If this information is available, it would also be interesting if the authors could comment upon this in relation to the findings of the satisfaction of the intervention.

  1. Is there any information available on average time/contact? (The range is described on page 5). Perhaps this is difficult to assess given the different type of contacts. If available it would be interesting to have this information when reading the discussion regarding the satisfaction and limitations of the intervention.

  1. 6.3 Would suggest the authors to consider moving section 2.6.3 to before 2.6.1 (to be new 2.6.1). This would also follow the order as it is presented in the results.

  1. The authors could consider moving the sentence “Data were analyzed from August 2020…” to the first sentence under the heading 2.7 alternatively to have it separated from the section on sample size.

  1. Would suggest the authors to consider using past tense for the entire analysis section. Currently there is a mix.

Results

  1. Heading 3.1 Should “family physician practice” be replaced with “PHC practice” for consistency?

  1. Have the authors considered presenting the intervention group before the control group in the tables (intervention group to the left)?
  2. Page 10. The sentence ending with “anxiety where the mean change in the control group might be better than in the intervention group”. Could this be reformulated to “anxiety where the mean decrease in anxiety in the control group might be larger than in the intervention group”

Discussion:

No additional comment to what has already been stated in introduction and methods.

References:

  1. General comment: Some journal names seem have their title words starting with capital letters and others not. Perhaps just double-check this for consistency.

  1. I believe ref nr 17 should be “world health organization.

  1. Ref nr 49 seem to be on a different position than the other references.

Reviewer 3 Report

Overall I felt that this paper had significant potential. How we support vulnerable individuals in the context of Covid-19 is obviously a matter of great applied importance. This obviously needs to be considered in terms of also protecting already stretched health resources. With that being said, there are some matters which I do feel need to be addressed. 

  • The manuscript needed more detail re what was involved in care as usual
  • I felt that the demographics and factors considered did not address a lot of variables which may relate to experiences of loneliness etc in the context of Covid-19 e.g. marital status, whether the individual lived alone, whether they were actually shielding due to either age or physical health difficulties, whether they were computer literate and the extent to which they are interacting with other individuals via online means. These factors would at very minimum need to be discussed. 
  • I would have welcomed some discussion of the changes in self ratings in the context of placebo effects.
  • The discussion says that the outcome measures may not have matched the intervention content. This confuses me slightly. I am wondering how this corresponds to the fact that self ratings did change. More discussion would be needed as to why some ratings changed and some did not. I do not feel that the current explanations have sufficient depth. 
  • On a related note to this, given that the appropriateness of the measures is used to some extent to explain non significant findings, I would have expected discussion of how the measures used here correspond to those used to evaluate other similar interventions- particularly those which have shown statistically significant effects.
  • Finally I feel that the conclusions need to be modified to reflect fact that intervention did not show statistically significant effects. I feel that the conclusions are too strongly worded given the results. 

I hope that you find this feedback useful.

Round 2

Reviewer 1 Report

Authors correctly addressed all my questions. Quality of the manuscript has significantly improved.